# Impact of Gender and Age on Rapid Eye Movement-Related Obstructive Sleep Apnea: A Clinical Study of 3234 Japanese OSA Patients

**DOI:** 10.3390/ijerph16061068

**Published:** 2019-03-25

**Authors:** Mamiko Mano, Tetsuro Hoshino, Ryujiro Sasanabe, Kenta Murotani, Atsuhiko Nomura, Reiko Hori, Noriyuki Konishi, Masayo Baku, Toshiaki Shiomi

**Affiliations:** 1Department of Sleep Medicine and Sleep Disorders Center, Aichi Medical University Hospital, 1-1 Nagakute, Aichi 4801195, Japan; mami0303@aichi-med-u.ac.jp (M.M.); hoshino.tetsurou.299@mail.aichi-med-u.ac.jp (T.H.); nomura.atsuhiko.689@mail.aichi-med-u.ac.jp (A.N.); rhori@aichi-med-u.ac.jp (R.H.); konishi.noriyuki.011@mail.aichi-med-u.ac.jp (N.K.); baku.masayo.859@mail.aichi-med-u.ac.jp (M.B.); toshiaki@aichi-med-u.ac.jp (T.S.); 2Biostatistics Center, Graduate School of Medicine, Kurume University, 67 Asahimachi, Kurume, Fukuoka 8300011, Japan; kmurotani@med.kurume-u.ac.jp

**Keywords:** obstructive sleep apnea, rapid eye movement-related obstructive sleep apnea, female sex hormone

## Abstract

Rapid eye movement (REM)-related obstructive sleep apnea (OSA) is characterized by apnea and hypopnea events due to airway collapse occurring predominantly or exclusively during REM sleep. Previous studies have reported that REM-related OSA occurs more commonly in women and younger individuals. However, external validity of this tendency has not been confirmed in a large clinical sample. The objective of this study was to evaluate the effect of gender and age on REM-related OSA after adjustment for several covariates based on their established clinical relationships to gender difference in OSA. A total of 3234 Japanese patients with OSA were enrolled in this study. We confirmed that female sex is an important risk factor for REM-related OSA, as reported by previous studies. Moreover, we showed that women aged over 50 years were at a greater risk than those aged under 50 years. These results suggest that hormonal changes in women might play an important role in REM-related OSA and might reflect its unknown pathophysiological characteristics.

## 1. Introduction

Obstructive sleep apnea (OSA) is a common disorder characterized by repetitive apnea and hypopnea due to complete or partial collapse of the upper airway during sleep despite ongoing breathing effort, resulting in oxygen desaturation, increased arousal from sleep, and excessive daytime sleepiness. The prevalence of OSA, defined as an apnea and hypopnea index (AHI) ≥ 5, is 22% in men and 17% in women, as shown by epidemiological studies [1]. The occurrence of OSA is strongly associated with cardiovascular diseases, including hypertension, heart failure, and stroke, and death [2,3,4,5]. In addition, several studies have reported that OSA is a potential risk factor for metabolic dysfunction, including obesity and type 2 diabetes [6,7].

Although upper airway collapse can occur during rapid eye movement (REM) and non-REM (NREM) sleep, withdrawal of excitatory noradrenergic and serotonergic inputs to the upper airway motor neurons during REM sleep further reduces upper airway dilator muscle activity and substantially increases the tendency of upper airway collapse [8]. Therefore, in the patients with OSA, REM sleep is strongly associated with increased frequency of respiratory events, including apnea and hypopnea, that are often prolonged and lead to severe desaturation.

REM-related OSA is the term characterized by apnea and hypopnea events due to airway collapse and occurs predominantly or exclusively during REM sleep [9]. Although the pathophysiological mechanism of REM-related OSA is unclear, clinically, it is important to classify OSA into non-stage specific and REM-related OSA because previous studies have indicated that OSA during REM sleep may contribute to a higher cardiovascular risk than OSA during NREM sleep [10]. Moreover, recent studies have reported that patients with REM-related OSA occasionally do not tolerate continuous positive airway pressure (CPAP) therapy, which is a common treatment option for OSA [11,12].

Previous studies have reported that REM-related OSA is more common in younger individuals and women [13,14], and the prevalence of REM-related OSA varies widely, ranging from 11.1% to 36.7% of all OSA cases, due to inconsistent definition of REM-related OSA [13,15,16,17]. However, the external validity of these tendencies has not been confirmed in a large clinical sample. Moreover, these reports, except that of Koo et al. [16], did not consider whether the pre- and postmenopausal status of women might have affected the association between REM-related OSA and female sex. In fact, several clinical studies have shown that reduction in the levels of female sex hormones is associated with increased OSA in women [18,19]. Therefore, we hypothesize that pre- and postmenopausal status might also affect the association between REM-related OSA and female sex.

The present study was conducted to confirm the external validity of the aforementioned characteristics of REM-related OSA, namely predominant occurrence in women and younger individuals, as reported in previous studies, and had two major objectives. First, we aimed to evaluate the gender difference in the prevalence of REM-related OSA in a large population of Japanese patients with OSA. Second, we aimed to clarify the association between REM-related OSA and female sex, considering aging-related hormonal changes.

## 2. Materials and Methods

### 2.1. Ethical Approval

This study is in compliance with the Declaration of Helsinki. The study was conducted along the ethical guidelines of Aichi Medical University Hospital, and the Ethical Committee of Aichi Medical University Hospital approved the study protocol before collection and analysis of the patient data (permission: 6/2016, approval number: 16-H038). Written informed consent was not obtained due to the retrospective nature of this study. Therefore, we disclosed the protocol of the study on the website, and subjects were offered the opportunity to opt out of the study.

### 2.2. Study Population

We assessed the medical records of the patients who were primarily diagnosed with OSA using nocturnal polysomnography (PSG) from March 2004 to April 2013. None of the patients had undergone surgical procedures, had been treated using CPAP therapy, or were using oral appliance. Patients aged < 20 and > 80 years were excluded from this study. Finally, 3234 patients were enrolled in this study. At the first visit, all patients were administered two questionnaires: the Epworth Sleepiness Scale (ESS), for measurement of subjective daytime sleepiness [20]; and the Self-Rating Depression Scale (SDS), for evaluation of depression status [21].

The ESS consists of 8 self-rated items which are scored from 0 to 3, and measure a subject’s daytime sleepiness in common situations of daily living; no time frame is specified. The ESS score is represented by the sum of individual item scores, and the total score ranges from 0 to 24. Values more than 10 are considered to indicate significant sleepiness.

The SDS consists of 20 self-rated items which are scored from 1 to 4, and measure a subject’s depressive symptoms. The SDS score is represented by the sum of individual item scores, and the total score ranges from 20 to 80. Values > 50 are considered to indicate depressive condition.

Abdominal, neck, and buttock circumferences, and body mass index (BMI) were also measured at the first visit.

### 2.3. Data Collection

Nocturnal PSG was performed using the Alice 4 or 5 system (Respironics, Inc., Murrysville, PA, USA). The following examinations were performed to continuously monitor biological variables: electroencephalography, bilateral electro-oculography, chin and anterior tibial electromyography, electrocardiography, airflow measurement using a nasal thermistor, measurement of respiratory effort based on thoracic and abdominal movements, body position, snoring sound, and arterial oxygen saturation. Apnea, hypopnea, and other PSG parameters were scored manually by sleep technicians according to the Rechtshaffen and Kales criteria [22]. Apnea was defined as a cessation of airflow for at least 10 s. Hypopnea was defined as a 50% reduction in airflow and/or respiratory effort, accompanied by oxygen desaturation of more than 3%, or arousal. The AHI was defined as the average number of apnea and hypopnea events per hour of sleep, and OSA was defined by AHI ≥ 5. The AHI was divided into AHI during REM sleep (AHI_REM_) and during NREM sleep (AHI_NREM_), and we evaluated the prevalence of REM-related OSA. According to previous reports, 3 criteria were used to determine REM-related OSA: definition #1: an overall AHI ≥ 5 and AHI_REM_/AHI_NREM_ ≥ 2; definition #2: an overall AHI ≥ 5, AHI_REM_/AHI_NREM_ ≥ 2, and AHI_NREM_ < 15; and definition #3: an overall AHI ≥ 5, AHI_REM_/AHI_NREM_ ≥ 2, AHI_NREM_ < 8, and REM sleep duration > 10.5 min [13,14].

All clinical records and PSG data were anonymized and grouped by gender. To test the effect of pre- and postmenopausal status on REM-related OSA, we set the threshold for menopausal status at 50 years of age, which was based on the mean menopausal age of 13,996 Japanese women [23]. Using these data, we evaluated the gender difference in the prevalence of REM-related OSA, and clarified the association between REM-related OSA and female sex with regard to pre- and postmenopausal status.

### 2.4. Statistical Analysis

Comparison between men and women in terms of baseline characteristics and PSG parameters were conducted using Mann-Whitney *U* test. Fisher’s exact test was conducted for categorical variables.

Logistic regression analysis was used to estimate the association between REM-related OSA and female sex, after adjusting for age under and over 50 years as categorical variables, BMI, cumulative percentage of time spent at oxygen saturation below 90% (CT90), AHI_REM_, and AHI_NREM_. The covariates were selected based on their established clinical relationships with gender difference in OSA [13,24,25,26,27]. Odds ratios have been presented, along with 95% confidence intervals.

All comparisons were two-tailed, and a *p* value < 0.05 was considered statistically significant. All data were analyzed using SAS software (version 9.4; SAS Institute, Inc., Cary, NC, USA).

## 3. Results

### 3.1. Baseline Characteristics and PSG Parameters in Male and Female OSA Patients

Table 1 shows the gender differences in the baseline characteristics and PSG parameters of OSA patients. The population of patients with OSA in this study predominantly included men, with a male-to-female ratio of 5.9:1; furthermore, the women were older (51.7 ± 13.5 vs. 57.1 ± 13.8 years, *p* < 0.001) and more obese (26.9 ± 5.0 vs. 27.2 ± 7.9 kg/m^2^, *p* = 0.008) than the men. Conversely, the average ESS score was significantly higher in men than in women (9.6 ± 5.6 vs. 8.2 ± 5.0, *p* < 0.001).

In the PSG parameters, we found significant differences between men and women in AHI (36.0 ± 24.0 vs. 28.1 ± 25.7, *p* < 0.001), AHI_NREM_ (35.4 ± 25.6 vs. 25.6 ± 27.5, *p* < 0.001), AHI_REM_/AHI_NREM_ (1.8 ± 3.1 vs. 4.3 ± 8.8, *p* < 0.001), mean respiratory event duration (26.4 ± 7.6 vs. 23.0 ± 8.9, *p* < 0.001), and CT90 (9.5 ± 16.5 vs. 6.4 ± 15.2, *p* < 0.001).

### 3.2. Gender Difference in the Prevalence of REM-Related OSA

Table 2 shows the prevalence of REM-related OSA in men and women. The prevalence of REM-related OSA according to definitions #1, #2, and #3 was 24.6%, 18.6%, and 12.2%, respectively, in the total study population.

When comparing patients with OSA aged under 50 years, the prevalence of REM-related OSA according to definitions #1, #2, and #3 was 22.8%, 16.5%, and 11.2% in men and 44.3%, 35.5% and 25.8% in women, respectively.

When comparing patients with OSA aged over 50 years, the prevalence of REM-related OSA according to definitions #1, #2, and #3 was 19.1%, 14.5%, and 8.7% in men and 47.7%, 37.5%, and 26.3% in women, respectively.

### 3.3. Association between REM-Related OSA and Sex in the Total Study Population

Table 3 shows the association between each definition of REM-related OSA and women in the total study population. Women were 3.212–3.342 times more likely to develop REM-related OSA than men in the unadjusted model. After adjusting for sex and age, in Model 1, women were 3.29–3.456 times more likely to develop REM-related OSA than men. After including the adjustments for Model 1 and adjusting for BMI, in Model 2, women were 3.512–3.646 times more likely to develop REM-related OSA than men. After including the adjustments for Model 2 and adjusting for CT90, in Model 3, women were 2.944–3.169 times more likely to develop REM-related OSA than men. After including the adjustments for Model 3 and adjusting for AHI_NREM_, in Model 4, women were 1.373–2.159 times more likely to develop REM-related OSA than men.

### 3.4. Association Between REM-Related OSA and Female Sex

#### 3.4.1. Association Between REM-Related OSA and Women Aged Under 50 Years

Table 4 shows the association between each definition of REM-related OSA and women aged under 50 years. In this analysis, 1259 male and 124 female patients with OSA aged under 50 years were included. Women were 2.643–2.687 times more likely to develop REM-related OSA than men in the unadjusted model. After adjusting for sex and BMI, in Model 1, women were 3.084–3.308 times more likely to develop REM-related OSA than men. After including the adjustments for Model 1 and adjusting for CT90, in Model 2, women were 2.116–2.33 times more likely to develop REM-related OSA than men. After including the adjustments for Model 2 and adjusting for AHI_NREM_, in Model 3, women were 1.192–1.419 times more likely to develop REM-related OSA than men.

#### 3.4.2. Association Between REM-related OSA and Women Aged Over 50 Years

Table 5 shows the association between each definition of REM-related OSA and women aged over 50 years. In this analysis, 1505 male and 346 female patients with OSA aged over 50 years were included. Women were 3.597–3.852 times more likely to develop REM-related OSA than men in the unadjusted model. After adjusting for sex and BMI, in Model 1, women were 3.576–3.866 times more likely to develop REM-related OSA than men. After including the adjustments for Model 1 and adjusting for CT90, in Model 2, women were 3.335–3.552 times more likely to develop REM-related OSA than men. After including the adjustments for Model 2 and adjusting for AHI_NREM_, in Model 3, women were 1.476–2.523 times more likely to develop REM-related OSA than men.

## 4. Discussion

To our knowledge, this is the first study to evaluate the association between REM-related OSA and female sex in a large population of Japanese patients with OSA with regard to pre- and postmenopausal status. We showed that AHI and AHI_NREM_ were higher in men than in women; however, there is no significant difference in AHI_REM_ between the two groups (Table 1). Similar results have also been reported by previous studies, and these findings have been explained by the clustering of apnea and hypopnea events during REM sleep in female patients [24,28]. In addition, the aforementioned result led to a significant difference in the AHI_REM_/AHI_NREM_ ratio in this study.

Previous studies have reported that the prevalence of REM-related OSA ranges from 11.1% to 36.7% of the patient population [13,14,15,16,17]. This variability is due to the use of inconsistent definitions of REM-related OSA and the small study population in some of the studies. The prevalence of REM-related OSA in our patient population ranged from 12.2% to 24.6%, depending on which of the three definitions were used (Table 2). These results are consistent with previous reports. We also evaluated the gender difference in the prevalence of REM-related OSA in patients aged under/over 50 years, and showed that REM-related OSA is more common in women, regardless of age. These tendencies of REM-related OSA are consistent with those in previous reports [24].

In logistic regression analysis, we selected age, BMI, CT90, and AHI_NREM_ as covariates, which have established clinical relationships with gender difference in patients with OSA [13,24,25,26,27]. After adjusting for these covariates, we confirmed that female sex is an important risk factor for REM-related OSA, as reported by previous studies (Table 3).

Previously, REM-related OSA was regarded to be more common in younger individuals [13,14,15,16]; however, these reports, except for Koo et al. [16], had not considered whether pre- and postmenopausal status might have affected the association between REM-related OSA and female sex. In this study, we showed that every adjusted and unadjusted odds ratio for REM-related OSA in the patients with OSA aged over 50 years was higher than that of patients aged under 50 years. These results were not consistent with previous reports and suggest that not only does female sex play a role in REM-related OSA, but pre- and postmenopausal status might contribute to its increased risk. In fact, several clinical studies have shown that reduction in the levels of female sex hormones is associated with increased OSA in women [18,19]. In addition, experimental models have shown that reduction in the levels of female sex hormones reduces the electromyographic activity of the genioglossus muscle [29,30], which is important to maintain pharyngeal patency. Moreover, reduction in genioglossus muscle activity may be an important contributor to REM-related OSA [31]. With respect to previous research, it can be assumed that reduction in the levels of female sex hormones cannot maintain pharyngeal patency, and this may lead to increase in the risk of REM-related OSA.

Our study has several limitations. First, we set the threshold for menopausal status at 50 years of age based on mean menopausal age of Japanese women [23]. However, none of the patients had been administered a questionnaire for menstrual history or tested for serum hormone levels. This may have led to a selection bias. Second, further selection bias may have been introduced by the fact this study was conducted in a single setting in Japan. Finally, the effects of unknown confounding factors, including usage of medication that affects REM sleep or decreases muscle tone, could not be excluded because of the retrospective nature of this study. Further studies are needed to confirm the external validity of our results.

## 5. Conclusions

To our knowledge, this is the first study demonstrating that female sex, beside age (over 50 years), is an important risk factor for REM-related OSA in the Japanese population. These results indicate that hormonal changes in women might play an important role in REM-related OSA and might reflect its unknown pathophysiological characteristics.

## Figures and Tables

**Table 1 ijerph-16-01068-t001:** Baseline characteristics and polysomnographic parameters of male and female OSA patients.

	Total (*n* = 3234)	Male (*n* = 2764)	Female (*n* = 470)	*p* Value
age (years)	52.5 ± 13.7	51.7 ± 13.5	57.1 ± 13.8	<0.001*
BMI (kg/m^2^)	27.0 ± 5.5	26.9 ± 5.0	27.2 ± 7.9	0.008 *
ESS	9.4 ± 5.5	9.6 ± 5.6	8.2 ± 5.0	<0.001 *
SDS	41.0 ± 9.2	41.0 ± 9.1	41.4 ± 9.8	0.82
TST (min)	396.5 ± 67.7	398.3 ± 66.6	385.5 ± 72.9	<0.001 *
Sleep Latency (min)	11.2 ± 17.4	10.7 ± 16.3	14.6 ± 22.3	<0.001 *
REM Latency (min)	120.6 ± 74.5	118.3 ± 73.0	134.0 ± 81.1	<0.001 *
REM sleep time/TST (%)	17.4 ± 6.9	17.4 ± 6.8	17.7 ± 7.2	0.43
Stage 1+2 sleep time/TST (%)	81.1 ± 8.4	81.4 ± 8.4	79.9 ± 8.5	<0.001 *
Stage 3+4 sleep time/TST (%)	1.5 ± 3.7	1.3 ± 8.53.6	2.5 ± 4.1	<0.001 *
AHI (events/h)	34.8 ± 24.4	36.0 ± 24	28.1 ± 25.7	<0.001 *
AHIREM (events/h)	36.7 ± 22.3	36.7 ± 22.0	36.8 ± 24.0	0.45
AHINREM (events/h)	34.0 ± 26.1	35.4 ± 25.6	25.6 ± 27.5	<0.001 *
AHIREM/AHINREM	2.2 ± 22.3	1.8 ± 3.1	4.3 ± 8.8	<0.001 *
Min SpO2 (%)	77.5 ± 12.3	77.4 ± 12.1	78.2 ± 13.8	0.001 *
Mean respiratory event duration (sec)	25.9 ± 7.9	26.4 ± 7.6	23.0 ± 8.9	<0.001 *
CT90(%)	9.1 ± 16.4	9.5 ± 16.5	6.4 ± 15.2	<0.001 *
Arousal Index (events/hr)	38.6 ± 21.8	39.8 ± 21.4	31.8 ± 22.9	<0.001 *

Continuous variables are expressed as mean ± standard deviation, and categorical variables are expressed as numbers (proportion). BMI, body mass index; ESS, Epworth Sleepiness Scale; SDS, Self-Rating Depressive Scale; TST, total sleep time; REM, rapid eye movement; NREM, non-rapid eye movement; AHI, apnea and hypopnea index; AHI_REM_, AHI during REM sleep; AHI_NREM_, AHI during NREM sleep; SpO_2_, peripheral capillary oxygen saturation; CT90, cumulative percentage of time spent at oxygen saturation below 90%. * *p* < 0.05.

**Table 2 ijerph-16-01068-t002:** Gender difference in the prevalence of REM-related OSA between age-matched groups.

	Total (n = 3234)	Under Aged 50			Over Aged 50		
		Male (n =1259)	Female (n = 124)	*p* Value	Male (n = 1505)	Female (n = 346)	*p* Value
REM-related OSA #1 (%)	796(24.6)	288(22.8)	55(44.3)	<0.001 *	288(19.1)	165(47.7)	<0.001 *
REM-related OSA #2 (%)	600(18.6)	208(16.5)	44(35.5)	<0.001 *	218(14.5)	130(37.5)	<0.001 *
REM-related OSA #3 (%)	394(12.2)	141(11.2)	32(25.8)	<0.001 *	131(8.7)	91(26.3)	<0.001 *

REM, rapid eye movement; OSA, obstructive sleep apnea. * *p* < 0.05.

**Table 3 ijerph-16-01068-t003:** Association between REM-related OSA and female sex.

	REM-Related OSA #1		REM-Related OSA #2		REM-Related OSA #3	
	OR	95% CI	OR	95% CI	OR	95% CI
Unadjusted	3.342	2.728–4.095	3.226	2.605–3.995	3.212	2.524–4.089
Model1	3.456	2.812–4.249	3.290	2.648–4.089	3.346	2.617–4.279
Model2	3.646	2.955–4.497	3.513	2.808–4.395	3.512	2.724–4.530
Model3	3.169	2.534–3.962	2.991	2.359–3.793	2.944	2.255–3.844
Model4	2.159	1.621–2.876	1.659	1.149–2.394	1.373	0.762–2.472

Dependent variables: 0 = non-stage specific OSA; 1 = REM-related OSA #1, REM-related OSA #2, and REM-related OSA #3. Model 1: adjusted for sex and age. Model 2: adjustments for Model 1 + adjusted for BMI. Model 3: adjustments for Model 2 + adjusted for CT90. Model 4: adjustments for Model 3 + adjusted for AHI_NREM_. OR, odds ratio; CI, confidence interval; BMI, body mass index; CT90, cumulative percentage of time spent at saturation below 90%; AHI, apnea and hypopnea index; REM, rapid eye movement; NREM, non-rapid eye movement; OSA, obstructive sleep apnea.

**Table 4 ijerph-16-01068-t004:** Association between REM-related OSA and women aged under 50 years.

	REM-Related OSA #1		REM-Related OSA #2		REM-Related OSA #3	
	OR	95% CI	OR	95% CI	OR	95% CI
Unadjusted	2.687	1.842–3.922	2.682	1.801–3.996	2.643	1.698–4.115
Model 1	3.084	2.086–4.559	3.308	2.171–5.040	3.197	2.006–5.094
Model 2	2.310	1.498–3.563	2.330	1.476–3.678	2.116	1.287–3.479
Model 3	1.419	0.810–2.488	1.335	0.633–2.814	1.192	0.357–3.981

Dependent variables: 0 = non-stage specific OSA; 1 = REM-related OSA #1, REM-related OSA #2, and. REM-related OSA #3. Model 1: adjusted for sex and BMI. Model 2: adjustments for Model 1 + adjusted for CT90. Model 3: adjustments for Model 2 + adjusted for AHI_NREM_. OR, odds ratio; CI, confidence interval; BMI, body mass index; CT90, cumulative percentage of time spent at saturation below 90%; AHI; apnea and hypopnea index; REM, rapid eye movement; NREM, non-rapid eye movement; OSA, obstructive sleep apnea.

**Table 5 ijerph-16-01068-t005:** Association between REM-related OSA and women aged over 50 years.

	REM-Related OSA #1		REM-Related OSA #2		REM-Related OSA #3	
	OR	95% CI	OR	95% CI	OR	95% CI
Unadjusted	3.852	3.009–4.931	3.597	2.773–4.667	3.744	2.776–5.048
Model 1	3.866	3.012–4.964	3.576	2.742–4.662	3.627	2.673–4.922
Model 2	3.552	2.732–4.617	3.335	2.518–4.417	3.413	2.477–4.703
Model 3	2.523	1.796–3.545	1.798	1.117–2.654	1.476	0.729–2.990

Dependent variables: 0 = non-stage specific OSA; 1 = REM-related OSA #1, REM-related OSA #2, and REM-related OSA #3. Model 1: adjusted for sex and BMI. Model 2: adjustments for Model 1 + adjusted for CT90. Model 3: adjustments for Model 2 + adjusted for AHI_NREM_. OR, odds ratio; CI, confidence interval; BMI, body mass index; CT90, cumulative percentage of time spent at saturation below 90%; AHI, apnea and hypopnea index; REM, rapid eye movement; NREM, non-rapid eye movement; OSA, obstructive sleep apnea.

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
