# Peer review of "Impact of Gender and Age on Rapid Eye Movement-Related Obstructive Sleep Apnea: A Clinical Study of 3234 Japanese OSA Patients"

_ijerph, 2019, doi:10.3390/ijerph16061068_

Round 1

Reviewer 1 Report

The authors have submitted a manuscript reporting the results of a retrosective study evaluating the relationship between the occurrence of REM-related OSA and certain demographic characterstics (sex, age). The topic is definitely worthy of research, the paper is well-written, the quality of English language used is excellent, and the overall approach is sound. I do have a few comments:

Major Comments

1. Under "Materials and Methods", "Study Population"; were there any exclusion criteria other than participant age? Were patients who were already using CPAP, who had undergone surgical procedures for OSA, or who were using dental devices for OSA excluded? Were patients with elevated central apnea indices excluded? Was a certain minimum sleep efficiency or percentage of REM sleep required for inclusion in the study?

2. The authors may consider exapanding on the Epworth Sleepiness Scale and Self-Rating Depression Scale; what types of questions are included, what the ranges of scores are, what the cutoff is for abnormality, what higher vs. lower scores mean, etc.

3. Have any other studies used the approach of grouping women into pre- and post-menopausal based on the national median age of menopause (in this case 50 years)? If so, the authors should cite them. I have my reservations about this approach, and attempts to draw conclusions about the relationships between menopausal states and other medical conditions based on it, unless the approach has been validated. The authors may consider eliminating this aspect of the study, and focus instead on the realtionship between gender and REM-related OSA. Alternatively, the authors may consider creating a smaller subset of patients who were most recently studied, administer (retrospectively but pertaining to their situation at teh time of their PSG) menopause-related questionnaires, and divide that subgroup into pre-menopausal and post-meonopausal for further analysis (including for the logistic regression analysis).

4. What statistical methods were used to obtain the data in Table 2?

5. Also, my understanding of Table 2 is that there are statistically significant differences between men under 50 and women under 50, and between men over 50 and women over 50, with regard to the prevalence of OSA. However, were women under 50 and women over 50 compared? This would appear to be an important aspect to evaluate (however see Major Comment #3). This also applies to Table 1.

6. I would like some clarification with regard to the independent variables chosen for the logistic regression tables presented in Tables 3-5. The dependent variable is the presence of REM-related OSA (which definition?). Age, gender and BMI are appropriate independent variables. However, what is the logic in including time spent with SpO2 below 90%, REM-AHI and NREM-AHI in evaluating the presence of a specific type of OSA, i.e., REM-related OSA? Also, why employ a stepwise model? My suggestion would be to perform a single logistic linear regression with presence of REM-specific OSA as the dependent variable and age, gender and BMI as independent variables and provide the results of the full anlaysis (with odds ratios for all the independent variables) so readers can see which of these independent variables are significant predictors.

7. Also, for Tables 4 and 5, see Major Comment #3. If the authors ultimately decide to use the approach of determining menopausal status based on a cutoff age of 50 years, then rather than perform two separate analyses for those two groups and compare the magnitude of the odds ratios (the approach that the authors have used but the validity of which is unclear to me), the authors may consider a single logistical regression analysis among women with age above or below 50 as a categorical variable and a proxy for menopausal status.

8. Page 7, line 228, the authors state, "In addition, the aforementioned results led to a significant difference in the AHI (REM)/AHI(NREM) ratio in this study". Is this a novel finding?

9. Page 8, line 245, the authors state, "These results suggest that not only does female sex play a role in REM-related OSA but pre- and post-menopausal status might constribute to its increased risk". Could the authors elaborate on this latter observation? Based on the statistical approach the authors have employed, it appears that the odds ratio for developing REM-related OSA among "postmenopausal" women is higher than in "pre-menopausal" women. Since REM-related OSA is considered a milder, and perhaps earlier, form of OSA, this would suggest that menopause is protective against OSA severity. This would run contrary to previously published work.      

Minor comments:

1. Page 3, line 89; the authors state, "despite arousal", I believe they meant "or arousal".
2. Page 3, line 110, typographical error ("date" should be "data").  
3. Heading for 3.3 should read "Association Between REM-related OSA and Sex" rather than "Association Between REM-related OSA and Women".
4. The authors may consider making the Results section more concise and avoiding bland repetition of data already presented in the tables in the text portion (Sections 3.1 through 3.42). 

Author Response

Dear reviewer1:

We are grateful for your time and constructive comments on our manuscript. We have implemented your comments and suggestions and wish to submit a revised version of the manuscript for further consideration in the International Journal of Environmental Research and Public Health . Changes made to the original manuscript are highlighted in the revised version. Below, we also provide a point-by-point response explaining our changes made in accordance with each of the editors or reviewers’ comments.

Thank you for your consideration. I lool forward to hearing from you.

Sincerely,

Tetsuro Hoshino

Department of Sleep Medicine and Sleep Disorders Center

Aichi Medical University Hospital, 1-1 Nagakute, Aichi, 4801195, Japan

Tel:81 561 62 3311; fax:81 561 62 4976

Email: hoshino.tetsurou.299@mail.aichi-med-u.ac.jp

Reviewer 2 Report

The present manuscript reports the impact of age and sex on REM-related OSA based on nocturnal polysomnography and self-reported questionnaires. This retrospective study is well written. I believe that it will be of interest to readership of IJERPH. However, there are some specific points that should be improved. Several concerns are mentioned below:

Introduction:

·         lines 56-61 are not completely clear and precise. The authors statement - “we hypothesize that pre- and postmenopausal status might also affect the association between REM-related OSA and female sex” don’t acknowledge comparable observations, although in a smaller samples, made by other groups. e.g.  Anttalainen et al., 2010 - “Impact of menopause on the manifestation and severity of sleep-disordered breathing”; Resta et al., 2003 - “Gender, age and menopause effects on the prevalence and the characteristics of obstructive sleep apnea in obesity”. The reference from the study (#16) Koo et al., 2008 also discuss the prevalence of REM related sleep-disordered breathing in women younger than 55 compared to those older than 55, a marker for menopausal status.

·         Lines 62-63 - “The present study was conducted to confirm the external validity of this disease characteristics reported in previous studies” - which characteristics, female predominance? Generalization to Japanese population or across populations?

Results/Discussion:

·         Lines 229-233 - The prevalence of REM-related OSA in the current study ranged from 12.2% to 24.6%; authors state that the prevalence rate is consistent with reported rates in previous studies that is 11.1% to 36.7%. I think that there is an apparent difference in REM-related OSA rate and authors should discuss this relatively lower prevalence.

·         There is some evidence that African-American women differ in sex hormonal status compared do Caucasian women. Is there any data about such differences in Japanese population?

·         The findings of the study indicate that “odds ratio for REM-related OSA in the patients with OSA aged over 50 years was higher than that of patients aged under 50 years”. Authors state that “these reports had not considered whether pre- and postmenopausal status” and refer to studies 13-16 (lines 241-242). However Reference #16 reports that REM SDB is more prevalent in women than in men and more prevalent in men and women younger than 55 than those older than 55. Different results from literature about REM related OSA prevalence and age /menopause in women should be discussed.

·          It has been reported that AHI of postmenopausal women is significantly greater than the AHI of women experiencing menses (Resta et al., 2003). Please report if there is a similar difference in your study - under and over of age 50. Similarly the difference in the depression score will be interesting. Women with OSA are reported to have depression more often than do men. It might explain the observed differences in the association of sex and REM-related OSA in women aged over and under 50y. Depending of your results it might be interesting to include depression in the Regression analyses.

Study limitations:

·         Study limitations should be extended. The gender imbalance (the participants in the study are mostly males) should be mentioned. How much your results might be related to the low percentage of women from the total sample.

Conclusion

·         Lines 265-266. Probably authors should add that this is a first study in the Japanese population.

Author Response

Dear reviewer2:

We are grateful for your time and constructive comments on our manuscript. We have implemented your comments and suggestions and wish to submit a revised version of the manuscript for further consideration in the International Journal of Environmental Research and Public Health . Changes made to the original manuscript are highlighted in the revised version. Below, we also provide a point-by-point response explaining our changes made in accordance with each of the editors or reviewers’ comments.

Thank you for your consideration. I look forward to hearing from you.

Sincerely,

Tetsuro Hoshino

Department of Sleep Medicine and Sleep Disorders Center

Aichi Medical University Hospital, 1-1 Nagakute, Aichi, 4801195, Japan

Tel:81 561 62 3311; fax:81 561 62 4976

Email: hoshino.tetsurou.299@mail.aichi-med-u.ac.jp

Round 2

Reviewer 1 Report

The authors have addresses the majority of my concerns. However, for Tables 4 and 5,the rationale they provide for not entering age below and above 50 years (as surrogate markers for pre and postmenopausal states) as categorical variables in a logistic regression analysis among female patients only remains unclear to me. They state that "the main objective of this study was simply to clarify the association between female sex and REM-related OSA, considering aging-related hormonal changes ". However, they did not study hormonal changes, and in the manuscript speak of pre- and post-menopausal status based on the mean cutoff of 50 years. Why then not use this as a categorical variable? it would seem the more statistically sound method.

Author Response

Dear Reviewer 1

We are grateful to the reviewer for their time and constructive comments on our manuscript.

We have implemented their commnents and suggestions and wish to submit arevised version of the manuscript for further consideration in the International Journal of Enviromental Research and Public Health. Changes made to the original manusript are highlighted in the revised revision. Below,we also provide a point-by-point respose explaing our changes made inaccordance with each of the reviewer’s comments.

Thank you for your consideration.I look forwasrd to hearing from you.

Sincerely,

Tetsuro Hoshino

Department of Sleep Medicine and Sleep Disorders Center

Aichi Medical University Hospital, 1-1 Nagakute, Aichi,4801195,Japan

Tel :+81 561 62 3311;fax;+81 561 62 4976

Email:hoshino.tetsurou.299@mail.aichi-med-u.ac.jp
